# A Computational Framework for Understanding the Impact of Prior Experiences on Pain Perception and Neuropathic Pain

**Malin Ramne** [1]*, **Jon Sensinger**[2,3]

**1** Department of Electrical Engineering, Chalmers University of Technology, Gothenburg, Sweden,
**2** Institute of Biomedical Engineering, University of New Brunswick, Fredericton, New Brunswick, Canada,
**3** Department of Electrical and Computer Engineering, University of New Brunswick, Fredericton, New Brunswick, Canada

\* malin@ramne.com

**Data Availability Statement:** Matlab code for all simulation results presented in this article and supporting information are available online at: https://zenodo.org/doi/10.5281/zenodo.10960405.

## Abstract

Pain perception is influenced not only by sensory input from afferent neurons but also by cognitive factors such as prior expectations. It has been suggested that overly precise priors may be a key contributing factor to chronic pain states such as neuropathic pain. However, it remains an open question how overly precise priors in favor of pain might arise. Here, we first verify that a Bayesian approach can describe how statistical integration of prior expectations and sensory input results in pain phenomena such as placebo hypoalgesia, nocebo hyperalgesia, chronic pain, and spontaneous neuropathic pain. Our results indicate that the value of the prior, which is determined by the internal model parameters, may be a key contributor to these phenomena. Next, we apply a hierarchical Bayesian approach to update the parameters of the internal model based on the difference between the predicted and the perceived pain, to reflect that people integrate prior experiences in their future expectations. In contrast with simpler approaches, this hierarchical model structure is able to show for placebo hypoalgesia and nocebo hyperalgesia how these phenomena can arise from prior experiences in the form of a classical conditioning procedure. We also demonstrate the phenomenon of offset analgesia, in which a disproportionally large pain decrease is obtained following a minor reduction in noxious stimulus intensity. Finally, we turn to simulations of neuropathic pain, where our hierarchical model corroborates that persistent non-neuropathic pain is a risk factor for developing neuropathic pain following denervation, and additionally offers an interesting prediction that complete absence of informative painful experiences could be a similar risk factor. Taken together, these results provide insight to how prior experiences may contribute to pain perception, in both experimental and neuropathic pain, which in turn might be informative for improving strategies of pain prevention and relief.

Data used to verify some model results are from Van Doorn, J., & Jepma, M. (2018, November 2): Behavioural and neural evidence for self-reinforcing expectancy effects on pain, available online at https://osf.io/bqkz3.

**Funding:** Funding for MR was provided by Promobilia Foundation (grant nr 19500). The funders had no role in study design, data collection and analysis, decision to publish, or preparation of the manuscript.

**Competing interests:** The authors declare that no competing interests exist.

## Author summary

To efficiently navigate the world and avoid harmful situations, it is beneficial to learn from prior pain experiences. This learning process typically results in certain contexts being associated with an expected level of pain, which subsequently influences pain perception. While this process of pain anticipation has evolved as a mechanism for avoiding harm, recent research indicates overly precise expectations of pain may in fact contribute to certain chronic pain conditions, in which pain persists even after tissue damage has healed, or even arises without any initiating injury. However, it remains an open question *how* prior experiences contribute to such overly precise expectations of pain. Here, we mathematically model the pain-learning-process. Our model successfully describes several counterintuitive but well-documented pain phenomena. We also make predictions of how prior experiences may contribute to the perception of pain and how the same learning process could be leveraged to improve strategies of pain prevention and relief.

## Introduction

Pain is beneficial for survival as it can motivate an organism to withdraw from harmful situations and avoid such harmful situations in the future. Despite its clear importance for survival the underlying mechanisms of pain are an active area of research. This limitation in knowledge, along with a wide variability in the pain experience, render many pain conditions difficult to treat. Chronic pain is one of the leading global burdens of health [1] as well as an enormous personal and economic burden, affecting more than 30% of people worldwide [2]. A first step toward understanding any complex phenomenon is formulating a precise definition. The International Association for the Study of Pain (IASP) have defined pain as [3]:

> *"An unpleasant sensory and emotional experience associated with, or resembling that associated with, actual or potential tissue damage"*

As a mechanism of protection from harm, pain should typically only arise when there is actual or potential tissue damage present, and the level of pain experienced should be proportional to the level of that tissue damage. However, pain can sometimes persist after the noxious stimulus has been removed or the tissue damage has healed. In fact, in some cases pain may arise even in absence of initiating tissue damage, as is the case with neuropathic pain (pain caused by a lesion or disease of the somatosensory nervous system [3]). Such maladaptive pain does not accurately reflect the level of harm to which the organism is exposed. How and why neuropathic pain may arise and persist in the absence of tissue damage is poorly understood, and consequently is often very difficult to treat.

Mathematical models of neuropathic pain have primarily focused on mechanisms in peripheral afferent neurons [4–6] and spinal cord neural circuits [6–8]. While neurophysiological changes at this level of the nervous system do contribute to neuropathic pain following peripheral nerve lesions, pain perception is also influenced by cognitive factors such as expectations based on previous experiences. It is suggested that this occurs through statistical integration of sensory input and prior information, for example through Bayesian inference [9–12]. An optimal estimate is obtained if the contribution of each source is determined by their respective level of uncertainty [13]. Bayesian inference has successfully been applied to describe a range of perceptual and sensorimotor tasks, both qualitatively and quantitatively [14–18], as well as pain phenomena such as placebo hypoalgesia and nocebo hyperalgesia

[19–22], and statistical pain learning [23] under experimental pain paradigms. As for chronic pain, Eckert et al., recently applied a Bayesian framework to illustrate how overly precise priors for being in a state of pain, combined with an ambiguous likelihood of perceiving pain from a sensory stimulation, increases the probability of transitioning to a chronic pain state [24]. Bayesian inference seems like a promising framework for modelling neuropathic pain, as damage to the nervous system often results in disrupted sensory input (corresponding to increased ambiguity of the likelihood), yet it remains an open question how overly precise priors in favor of pain might arise.

How the prior distribution is defined plays an important role in Bayesian inference [13]. One common Bayesian approach for tracking a state that may change over time is Kalman filtering [25], in which the prior is produced by an internal model of the world [13,25]. Here, we first verify that Kalman filtering can describe how statistical integration of prior expectations and sensory input results in placebo hypoalgesia and nocebo hyperalgesia, as demonstrated through experiments. Through simulations we also explore how the same mechanisms can contribute to chronic pain and spontaneous neuropathic pain. Next, to address how priors can be influenced by previous experiences of pain, we present a modified Kalman filter model of pain perception, adapted from previously published models on motor adaptation [15–18]. In this model the state estimate (pain) is computed using the standard Kalman algorithm. A second Kalman filter is used to update the parameters of the internal model in the first Kalman filter, based on the difference between the predicted and the perceived pain. This hierarchical structure allows us to model how the context and dynamics of previous pain experiences can give rise to and shape future expectations. Our model simulations predict that the dynamics of previous experiences of pain and stochastic fluctuations of the parameters in the internal model, coupled with ambiguity in sensory input following damage to the nervous system, may be key drivers of neuropathic pain.

## Results

### Single-layer Kalman filter

Pain is dynamic, meaning that the intensity, quality, and other characteristics of the experience change over time. One common Bayesian approach for modelling such temporal dynamics is Kalman filtering [25]. In the Kalman filter a state, $x$, is estimated based on sensory input and a prior estimate, which is produced by an internal model of the world [13,25]. In the context of pain, we suggest that the state $x$ represents the true level of actual or potential tissue damage (hereafter referred to as *tissue damage* for brevity), and the estimate $\hat{x}$ represents the perceived pain. The tissue damage gives rise to sensory input, $z$, for example in the form of activation of nociceptive afferent neurons. In our model, the prior estimate, $\hat{\bar{x}}$, correspond to the expected level of pain. At each time point $k$, we assume that there are two main factors contributing to the prior: the level of perceived pain in the previous time step, $\hat{x}^{(k-1)}$, and the control input, $u^{(k)}$, which corresponds to factors that may predict changes in the perceived level of pain, such as motor actions, environmental context and cues, and possibly even psychological processes such as thoughts and emotions [26]. For ease of communication, we use linear concepts in this model, such that $\hat{\bar{x}}^{(k)} \propto \hat{A}\hat{x}^{(k-1)} + \hat{B}u^{(k)}$ where $\hat{A}$ and $\hat{B}$ are the parameters of the internal model, but the ideas we are exploring extend also to nonlinear concepts. In this first part we will start by showing that, similar to other Bayesian models explored in literature [19,20,24,27], this Kalman filter can successfully model a variety of pain phenomena, including placebo hypoalgesia, nocebo hyperalgesia, and chronic pain and additionally can produce dynamics representative of spontaneous neuropathic pain.

The influence of expectations on pain perception is commonly and perhaps most clearly demonstrated by the phenomenon of placebo hypoalgesia (or the reverse effect, nocebo hyperalgesia), where identical noxious stimuli may be perceived as less (more) painful depending on the expectations associated with the context in which the stimuli is delivered [28]. Classical conditioning is a procedure that is often used to induce placebo or nocebo responses in experimental studies [29]. In the conditioning procedure a conditioned stimulus (previously neutral stimulus, e.g., a visual cue) is repeatedly paired with an unconditioned stimulus (e.g., a noxious stimulus). Through this process, the conditioned stimulus can come to elicit a response originally associated with the unconditioned stimulus, a so-called conditioned response. In the context of placebo hypoalgesia and nocebo hyperalgesia, the conditioned response is an expectation of experiencing low or high pain upon presentation of the conditioned stimuli. We simulate the placebo and nocebo effect resulting from such a classical conditioning procedure by letting the conditioned stimulus correspond to the control input, $\boldsymbol{u}^{(k)}$. We let the control input have identical magnitude in placebo and nocebo trials, to reflect that the cue that is used as the conditioned stimulus does not inherently signal different levels of pain. The placebo and nocebo effects arise from the elements in the internal model parameter $\hat{B}$ having different values for the cues associated with high or low pain, such that $\hat{b}_{placebo} < \hat{b}_{nocebo}$. Our simulation results are presented in Fig 1, along with experimental data from an open-source dataset provided by Jepma et al., [20].

It is also suggested that the effects of expectations can extend beyond acute pain and could contribute to pain persisting even after the noxious stimulus has been removed and tissue damage has healed [9–12]. In our model this disproportionate pain results from a mismatch between the internal model parameter $\hat{A}$ and the real-world rate of tissue recovery. The contribution of the expectation and the sensory input to the final estimate is determined by their respective uncertainty. The uncertainty in sensory input is denoted by $R$, where larger values of $R$ indicate more uncertainty. The simulation results presented in Fig 2 demonstrate elevated uncertainty in the sensory input (higher value of $R$), in combination with the internal model parameter $\hat{A} = 1$ results in the perceived pain being significantly higher than the true level of tissue damage and persisting even when the tissue damage has recovered.

In the context of Bayesian models, the more uncertainty there is in the likelihood and the less uncertainty there is in the prior, the more the posterior will correspond to the prior. In our model, this means that the perceived level of pain will be strongly influenced by the expected pain if and only if there is more uncertainty in the sensory input than in the expectation. We assume that, at baseline, the uncertainty in sensory input is low, but that e.g., tissue damage can cause some disturbance to the sensory receptors that innervate the affected area, resulting in increased uncertainty in sensory input relating to that region. Sensory input may also be affected by damage directly to peripheral afferent nerves or regions of the central nervous system involved in processing of sensory input. Under these circumstances, pain may arise spontaneously, even in the absence tissue damage. This type of pain is commonly referred to as neuropathic pain. The gradual increase of pain before the onset of tissue damage in the bottom right panel of Fig 2 is one example of how pain can arise spontaneously when the uncertainty in the sensory input is elevated. In Fig 3 we provide additional examples of how, in our model, the expected level of spontaneous neuropathic pain following for example a nerve injury depends on the level of uncertainty in the sensory input (value of $R$) and the value of the internal model parameter $\hat{A}$.

By modeling pain perception as a Kalman filter our results so far have successfully demonstrated pain phenomena such as placebo hypoalgesia, nocebo hyperalgesia, chronic pain, and spontaneous neuropathic pain. We have shown that the level of uncertainty in the sensory

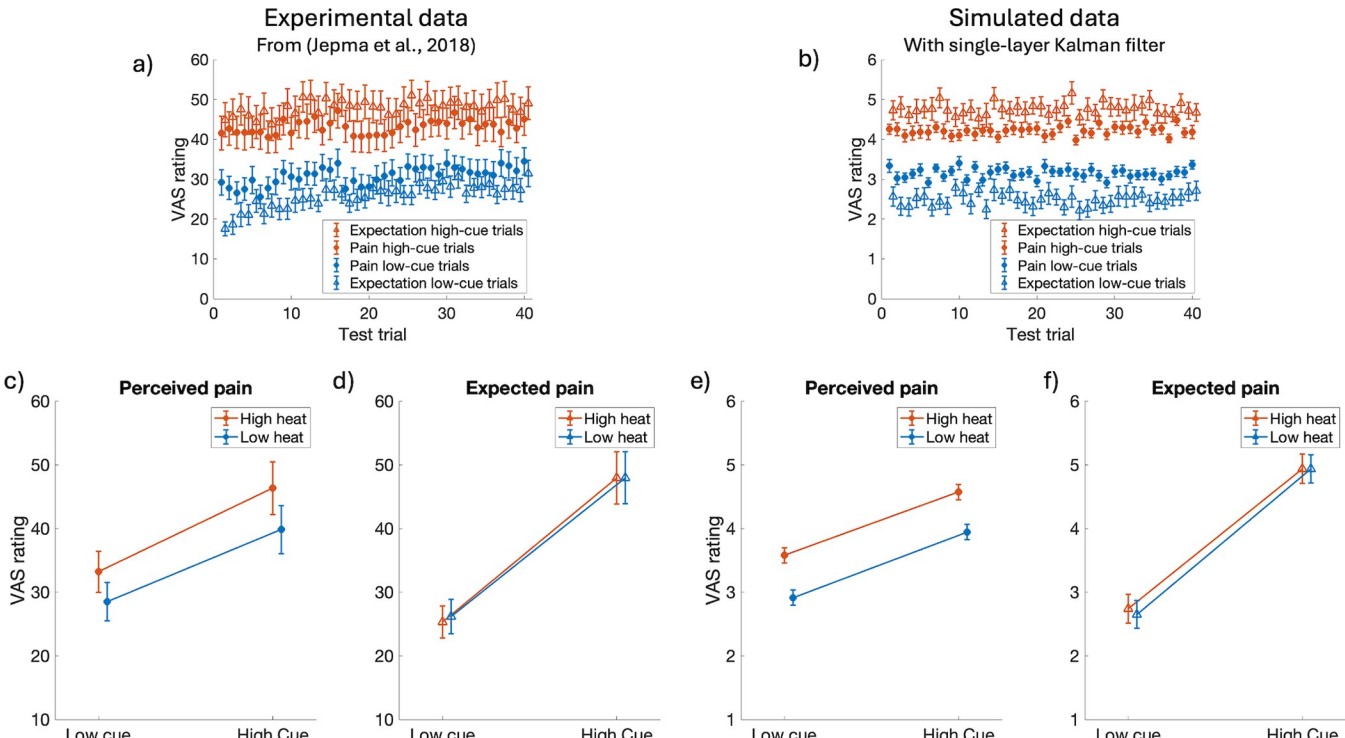

**Fig 1.** Experimental data from Jepma et al., [20] (left) and our Kalman filter simulations results (right). Placebo hypoalgesia (or the reverse effect, nocebo hyperalgesia) is the phenomenon where identical noxious stimuli may result in lower (higher) perceived pain ($\hat{x}$, filled circles) depending on expectations ($\hat{\hat{x}}$, open triangles). In classical conditioning the expectations are influenced by cues that have been associated with high or low noxious stimuli ('high cue' and 'low cue'). In testing the effect of conditioning Jepma et al., applied two different levels of noxious heat stimuli ('low heat' and 'high heat'). Note that the during the test trials level of noxious stimuli is independent from the cue, i.e., the low cue is paired with both high and low thermal stimulation. **a)** and **b)** average expected (open triangles) and perceived (filled circles) pain as a function of cue type and trial for experimental and simulated data, respectively. **c)** and **e)** perceived (filled circles,), **d)** and **f)** expected (open triangles) pain as a function of stimulus temperature and cue type for experimental and simulated data, respectively. Error bars indicate inter-individual standard errors. Note that the experimental data is measured on a 100-unit scale, whereas the simulated data is on an 11-unit scale.

input and the values of the internal model parameters may be key contributors to these phenomena. What we have not been able to account for with this model is *how* the values of the parameters in the internal model are determined. The internal model parameters are typically tuned by hand or fit to experimental data (e.g., [19–22]). Although these manually tuned models do give insight to how expectations can influence pain perception, they do not account for how the dynamics of previous pain experiences give rise to those expectations. To incorporate how previous experiences contribute to pain expectations we must turn to descriptive models of how the internal model is updated.

## The Hierarchical Kalman filter

The traditional Kalman filter gives us a framework for understanding how loss of certainty in sensory input, combined with predictions from an internal model of the world, might contribute to pain phenomena such as placebo hypoalgesia, nocebo hyperalgesia and spontaneous neuropathic pain. However, in this framework the internal model parameters are manually tuned, and therefore the traditional Kalman filter alone cannot account for how previous experiences influence the internal model and the resulting pain expectations. To address this shortcoming, we turn to recent work on modelling motor adaptation, where a second Kalman filter

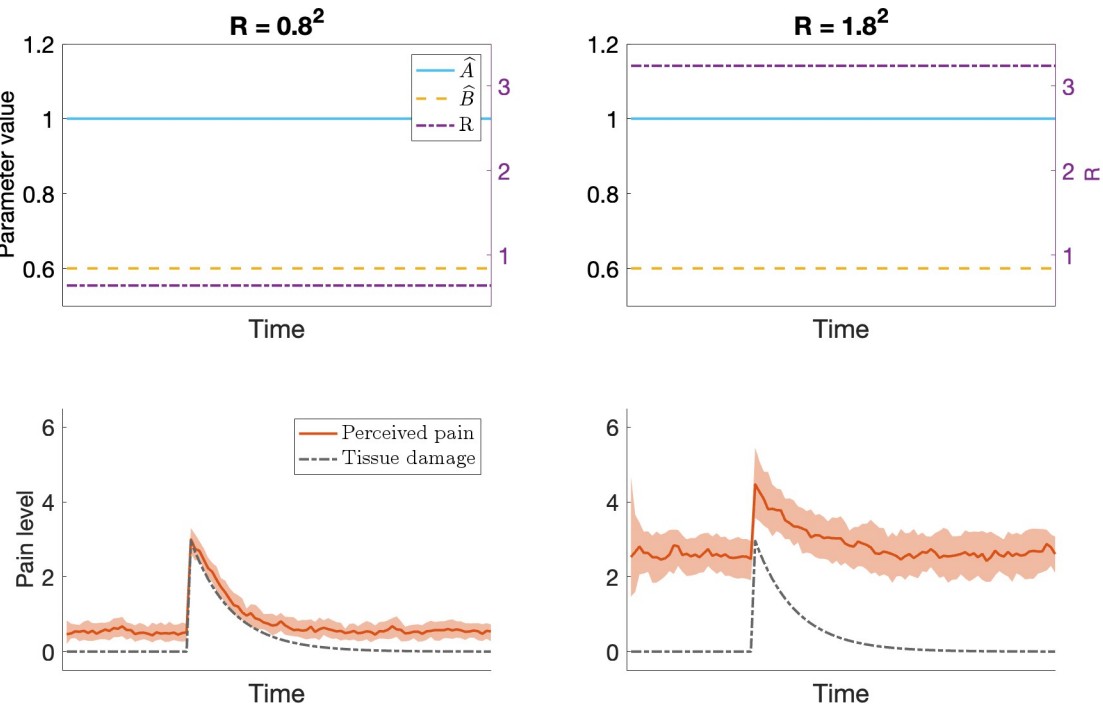

**Fig 2. Results of the Kalman filter simulations of chronic pain.** The model produces output reflecting chronic pain when there is elevated uncertainty in the sensory input, in combination with the internal model parameter $\hat{A} \approx 1$ (solid blue line in upper plots). Left: when R (dash-dotted, purple line in upper plot) is low the perceived pain, $\hat{x}$, (solid red line in lower plot, the shaded area indicates the interquartile range) will primarily be influenced by the sensory input and closely correspond to the true level of tissue damage, $x$, (dash-dotted black line in lower plot).Right: for a larger value of R, predictions from the internal model have a stronger influence on the perceived level of pain, possibly resulting in a chronically elevated level of pain even after the tissue damage has recovered.

has been used to update the parameters of the internal model based on the difference between the prediction and the posterior estimate [15–18]. This process reflects that people integrate previous experiences in their future expectations. Here, we adapt the hierarchical Kalman filter to the context of pain and demonstrate with an example of classical conditioning that the model successfully reflects how previous pain experiences influence future expectations.

To verify that the hierarchical model reproduces similar results as the single-layer Kalman filter regarding placebo hypoalgesia and nocebo hyperalgesia, we turn to the concept of classical conditioning. In Fig 4 we demonstrate the evolution of the internal model parameter $\hat{B} = [\hat{b}_{placebo}, \hat{b}_{nocebo}]$ and the expected pain across the conditioning trials in panels a) and b), along with the results of the simulated placebo and nocebo conditioning in panels c) and d). Inspection of panel a) reveals that at baseline (the first conditioning trial) the distribution of the values of $\hat{b}_{placebo}$ and $\hat{b}_{nocebo}$ overlap, resulting in no difference in the expected pain associated with the high and low cues (panel b)). Across the conditioning trials, the values of $\hat{b}_{placebo}$ and $\hat{b}_{nocebo}$ start differentiating, such that $\hat{b}_{placebo} < \hat{b}_{nocebo}$, creating expectation of low pain associated with the low cue, and high pain associated with the high cue. Finally, the effect of the conditioning is tested by pairing the conditioned stimuli with intermediate-intensity unconditioned stimuli (47˚C for low heat, 48˚C for high heat, independent of the visual cue), revealing that the simulated conditioning procedure results in placebo hypoalgesia and nocebo hyperalgesia that

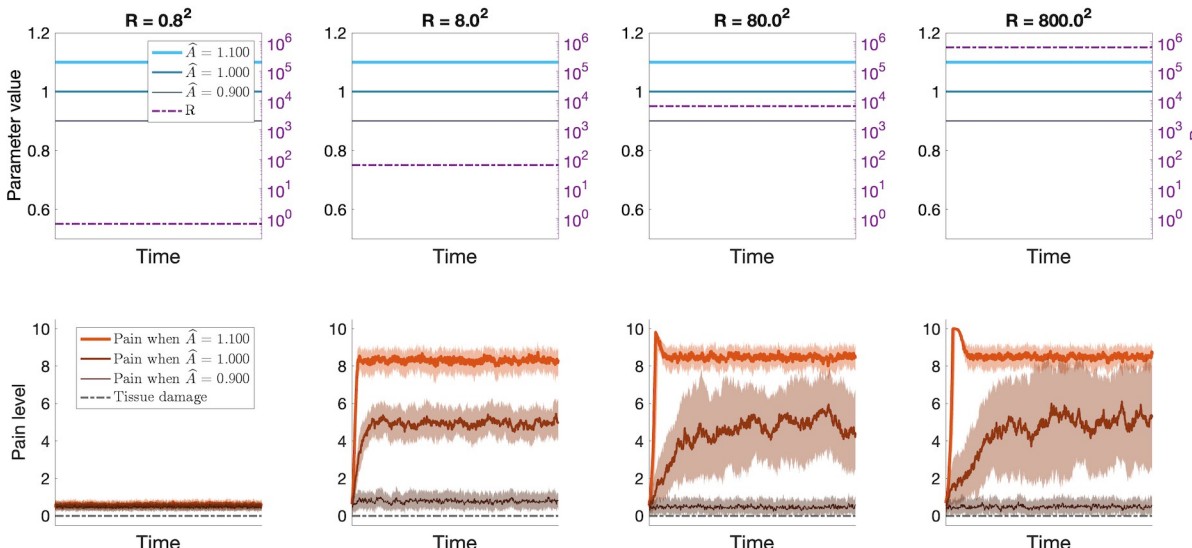

**Fig 3. Results from Kalman filter simulations of neuropathic pain.** Damage to the sensory nervous system may result in increased uncertainty of sensory input relating to the level of tissue damage. Depending on the level of uncertainty (value of $R$, dash-dotted purple line in upper plots) and the value of the internal model parameter $\hat{A}$ (solid blue lines in upper plots), these changes could result in spontaneous neuropathic pain (solid red lines in lower plot, the shaded area indicates the interquartile range).The initial overshoot for $\hat{A} = 1.1$ in the two rightmost plots, where $R = 80^2$ and $R = 800^2$, is caused by the initial variance of the estimate $P$, being low. When $P \ll R$, the posterior estimate is dominated by the prior, which here predicts increasing pain since $\hat{A} > 1$. Due to the uncertainty in the sensory input, the value of $P$ rapidly increases, until an equilibrium is reached, in which the posterior still is strongly influenced by the prior, but also has some influence from the noisy sensory input.

qualitatively matches the experimental data and the pattern produced by the single-layer Kalman filter in Fig 1.

In the single-layer Kalman filter we noted that the value of the internal model parameter $\hat{A}$ and the uncertainty in the current sensory input, $R$, influence the level of spontaneous neuropathic pain following nerve injury (Fig 3). In the hierarchical Kalman filter model, the value of $\hat{A}$ changes over time, which affects the level of predicted pain. Furthermore, the value of $R$ influences how the internal model updates. These differences result in pain dynamics that differ from what was observed in the single-layer Kalman filter. In particular, the model predicts that the value of $\hat{A}$ at the time of a nerve injury and the level of sensory disruption (i.e., value of $R$) may play a role in the level and variance of subsequent neuropathic pain, see Fig 5.

The classical conditioning example demonstrated how repeated painful experiences allow the internal model to update over time to give more accurate predictions of the real world. Later, we saw that the value of $\hat{A}$ at the time of a nerve injury may determine the existence and level of neuropathic pain. Taken together, the same processes as in classical conditioning could occur also in other contexts, and result in values of the internal model parameters that predispose certain individuals to neuropathic pain. In the hierarchical Kalman filter the value of $\hat{A}$ is determined by previous painful experiences. Intuitively, exposure to persistent pain ($A \approx 1$) typically results in values of $\hat{A}$ close to 1, while exposure to quickly transient pain ($A < 1$) typically gives lower values of $\hat{A}$.

But what happens if there is no exposure to painful experiences? Since there are no periods of decaying pain to inform the internal model, a tonically pain-free state also results in $\hat{A} \approx 1$, regardless of the true value of $A$. Fig 6 shows how persistent pain (left), quickly transient pain (middle) and complete absence of pain (right) each influence the internal model, and the

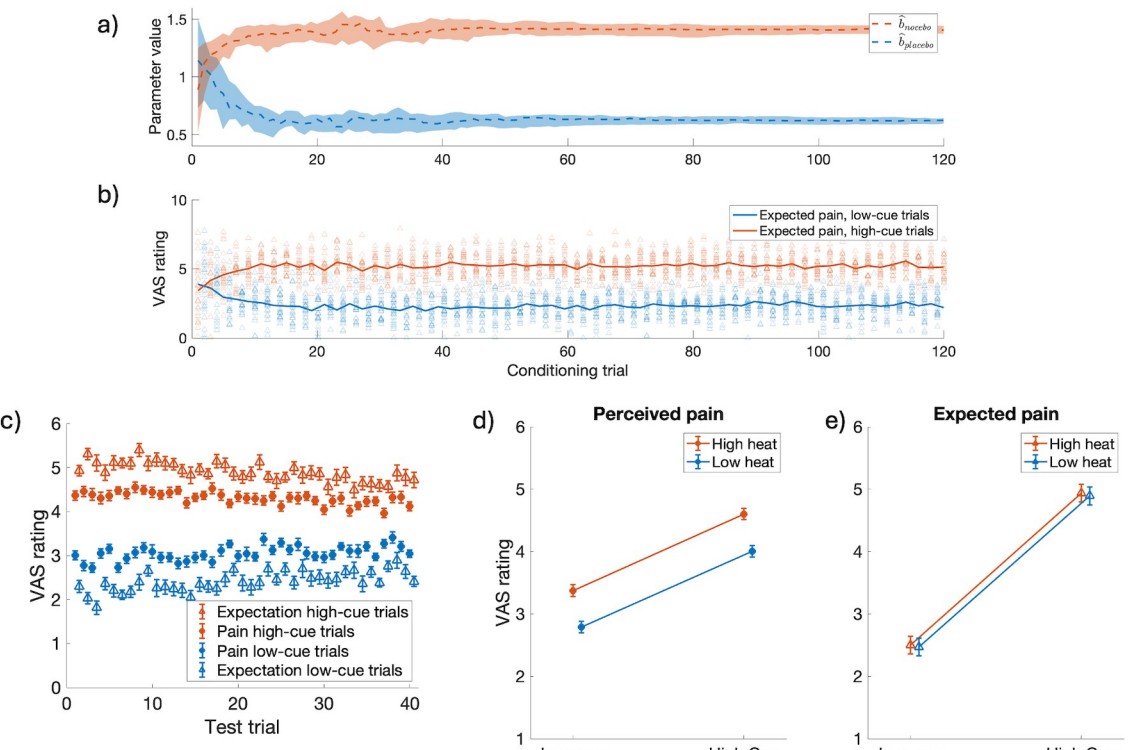

**Fig 4. Results of the hierarchical Kalman filter simulations of classical conditioning.** We simulate the conditioning procedure similar to the learning phase of the experimental paradigm described by Jepma et al., [20]. During conditioning previously neutral cues ('high cue' and 'low cue' in the figures) are repeatedly paired with high or low noxious stimuli, resulting in diverging values of internal model parameters $\hat{b}_{placebo}$ and $\hat{b}_{nocebo}$ (panel **a**)), and creating expectations of high pain associated with the high cue, and low pain associated with the low cue (panel **b**)). In testing the effect of conditioning the cues are paired with intermediate intensity noxious stimuli (47˚C for 'low heat' and 48˚C for 'high heat'). Note that the during the test trials the level of noxious stimuli is independent from the cue, i.e., the low cue is paired with both high and low thermal stimulation. **a**) median value of $\hat{b}_{nocebo}$ (dashed red line) and $\hat{b}_{placebo}$ (dashed blue line) across conditioning trials. Shaded areas indicate the interquartile range. **b**) average expected pain for high-cue trials (red) and low-cue trials (blue) during conditioning. Open triangles indicate the expected pain for each participant on each conditioning trial. **c**) the average expected ($\hat{\hat{x}}$, open triangles) and perceived ($\hat{x}$, filled circles) pain as a function of cue type on each test trial. **d**) perceived (filled circles) and **e**) expected (open triangles) pain as a function of stimulus temperature and cue type. Error bars indicate inter-individual standard errors.

resulting level of neuropathic pain following a nerve injury (vertical dashed line). As we will elaborate in the discussion section, our model offers a promising framework to investigate and model how persistent non-neuropathic pain may be a risk factor for developing neuropathic pain following denervation [30–32], and additionally offers an interesting prediction that complete absence of informative painful experiences could be a similar risk factor.

Another phenomenon that relates to the temporal dynamics of pain perception is *offset analgesia*, which is defined as a disproportionally large pain decrease after a minor noxious stimulus intensity reduction [33]. The phenomenon can be elicited by a simple experimental paradigm: noxious thermal stimuli are applied to an area of skin in three consecutive time intervals T1, T2 and T3, where the same temperature is applied during T1 and T3, and the temperature during T2 is slightly higher. This stimulation paradigm typically results in a disproportionate pain reduction in response to the temperature reduction from T2 to T3 as compared to a control condition where a constant temperature is applied throughout all three time intervals. We were curious to see if the adaptive nature of the hierarchical Kalman filter would

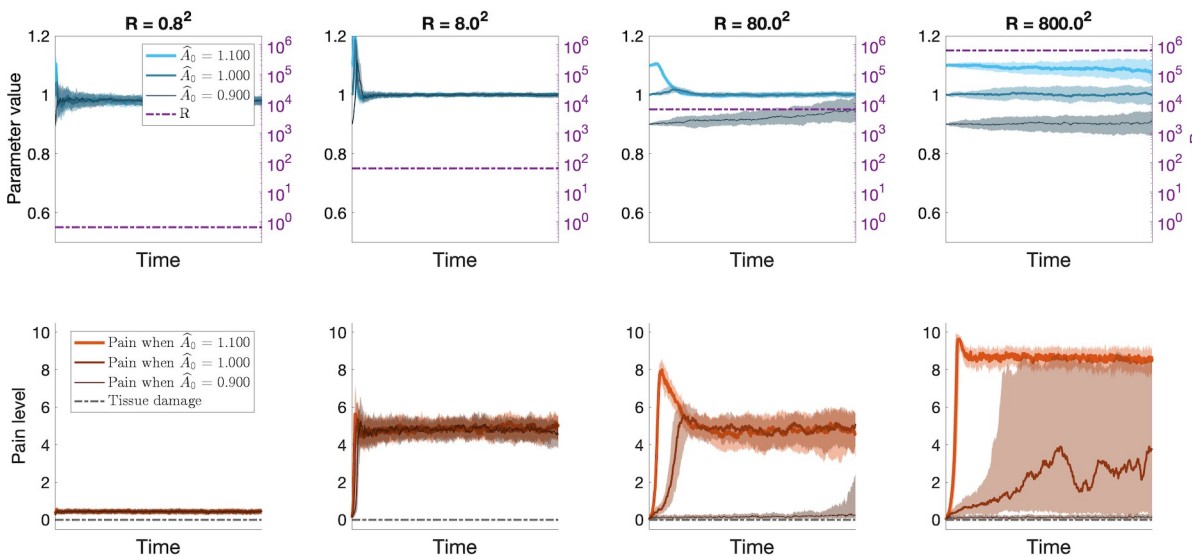

**Fig 5. Results from hierarchical Kalman filter simulations of how the value of $\hat{A}$ (solid blue lines in upper plots) at the time of a nerve injury and the level of sensory disruption (i.e., value of $R$, dash-dotted purple line in upper plot) may play a role in the characteristics of subsequent neuropathic pain.** Simulations were run for three different initial values of $\hat{A}$ : $\hat{A}_0 = \{0.9, 1.0, 1.1\}$. The top panels show the inferred value of $\hat{A}$ over time, and the bottom panels show the corresponding level of perceived pain, $\hat{x}$. For $R = 0.8^2$ (left), the perceived pain (solid red lines in the lower plots) is strongly influenced by sensory input and remains close to 0. For $R = 8^2$ (second from the left), $\hat{A} \approx 1$ regardless of the initial value, and the perceived pain is tonically at an intermediate intensity with little variance. For $R = 800^2$ (right), the value of $\hat{A}$ is no longer at all influenced by sensory input and reduces to a random walk centered at the initial value $\hat{A}_0$ and variance equal to the noise in the internal model. In this scenario, if $\hat{A}_0 = 1.1 > 1$ the pain is likely to stay tonically high, and similarly if $\hat{A}_0 = 0.9 < 1$ the level of pain is likely to drop to stay at 0. If $\hat{A}_0 \approx 1$, the perceived level of pain may vary widely as the value of $\hat{A}$ fluctuates above and below 0. For $R = 80^2$ (second from the right) the model displays a mixture of the behavior described for $R = 8^2$ and $R = 800^2$. The shaded areas indicate the interquartile ranges.

enable the model to reproduce this phenomenon. As can be seen in Fig 5, the model produces an underdamped response following a reduction in noxious stimuli. This effect is qualitatively similar yet smaller than what is typically observed in experimental studies of offset analgesia. We will elaborate on possible reasons for the magnitude discrepancy in the discussion section.

## Discussion

Here, we first verified that the Kalman filter, a common Bayesian approach for modelling temporal dynamics, produced results in line with experimental data and previous, similar Bayesian models, and additionally offered insight to how disruption of sensory input may contribute to spontaneous neuropathic pain. Next, to address how previous experiences contribute to pain expectations we applied a second Kalman filter to update the parameters of the internal model based on the difference between the prediction and the posterior estimate. With this hierarchical model structure, we could again model placebo hypoalgesia and nocebo hyperalgesia that qualitatively matches results from previous Bayesian models, but also show how these phenomena can arise from previous experiences in the form of a classical conditioning procedure. We also showed that, due to the adaptive nature of the hierarchical Kalman filter, the model produced a disproportionally large pain decrease after a minor noxious stimulus intensity reduction, a phenomenon that is commonly referred to as offset analgesia. Taken together, these results indicate that the hierarchical Kalman filter provides a promising framework for modelling how pain inference may be influenced by the context and dynamics of previous pain experiences.

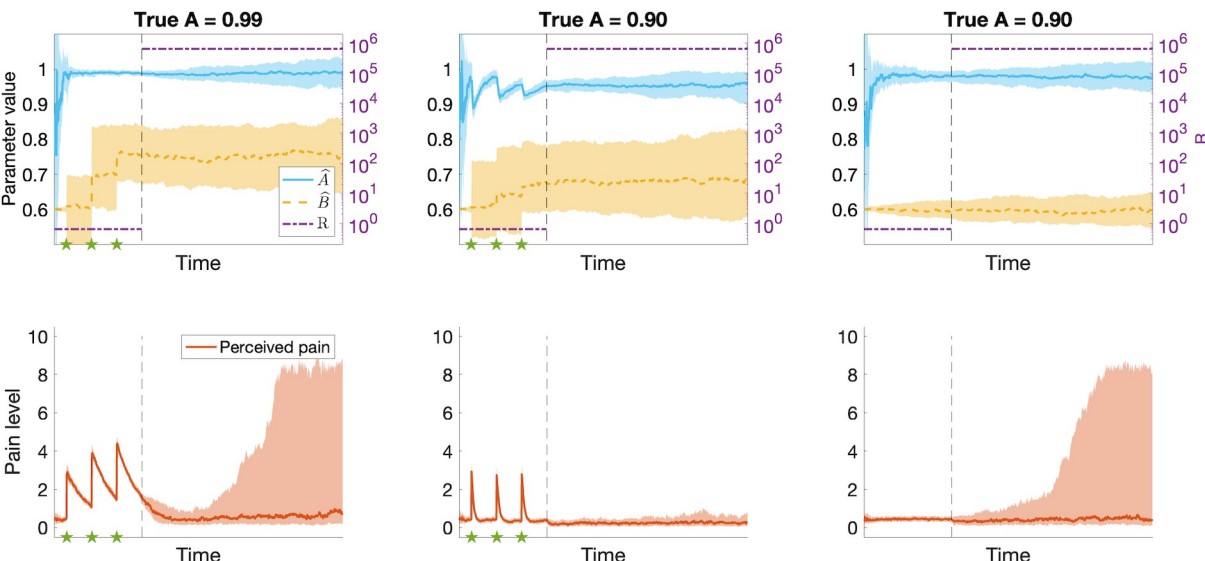

**Fig 6. Results from hierarchical Kalman filter simulations of how the value of the internal model parameters are determined by previous painful experiences and may contribute to neuropathic pain following a nerve injury.** In the left panels, $A$ = 0.99, yielding persistent pain (solid red line in lower plots) following noxious stimuli (indicated by stars). This value of $A$ results in $\hat{A} \approx 1$ (solid blue line in the upper plot), and a high risk of spontaneous neuropathic pain following a nerve injury (corresponding to a change in the value of R, indicated by the vertical dashed line). In the middle panels $A$ = 0.9, resulting in quickly transient pain, a lower value of $\hat{A}$ and a lower risk of spontaneous neuropathic pain. In the right panels $A$ = 0.9 again, but in this simulation, there are no noxious stimuli. These circumstances result in $\hat{A} \approx 1$ and a higher risk of spontaneous neuropathic pain than in the example in the middle. The shaded areas indicate the interquartile ranges.

In the proposed modelling framework, spontaneous neuropathic pain will arise if the following two criteria are fulfilled: 1) the internal model predicts pain, and 2) the uncertainty in the sensory input, denoted $R$, is large. With the hierarchical Kalman filter we could explore how the context and dynamics of previous pain experiences could result in the internal model predicting pain. Our simulations corroborated that persistent non-neuropathic pain is a risk factor for developing neuropathic pain following denervation, which is well established in previous literature [30–32] and additionally offered an interesting prediction that complete absence of informative painful experiences could be a similar risk factor. Neuropathic pain is often resistant to common pharmacological pain treatments [34,35]. Therefore, finding alternative ways of relieving or preventing neuropathic pain is of utmost interest. Our simulations provide a hint at possible preventative measures, such as avoiding persistent pain before denervation. This is in line with existing literature on risk factors for developing neuropathic pain [30–32]. Another prediction from the model, which might seem controversial, is that exposure to transient pain before denervation might be another strategy for mitigating the risk of developing neuropathic pain. This prediction remains to be established empirically.

When it comes to relieving existing neuropathic pain, we turn to the second criteria for neuropathic pain; the uncertainty in the sensory input, $R$. Input relating to tissue injury could come in several different sensory modalities, such as nociceptive, mechanoreceptive and thermoceptive afferent input. After nerve injury some (or all, e.g., following amputation) of these sensory modalities could be affected, resulting in increased uncertainty of the sensory input. Defining specific sensory modalities could be an interesting avenue for future development of the model, as it could provide insight to how central processing might contribute to pain conditions such as allodynia, a pain phenomenon where pain is elicited by otherwise non-painful stimulation such as cold, warmth or touch. As for relieving existing neuropathic pain, if $R$ can

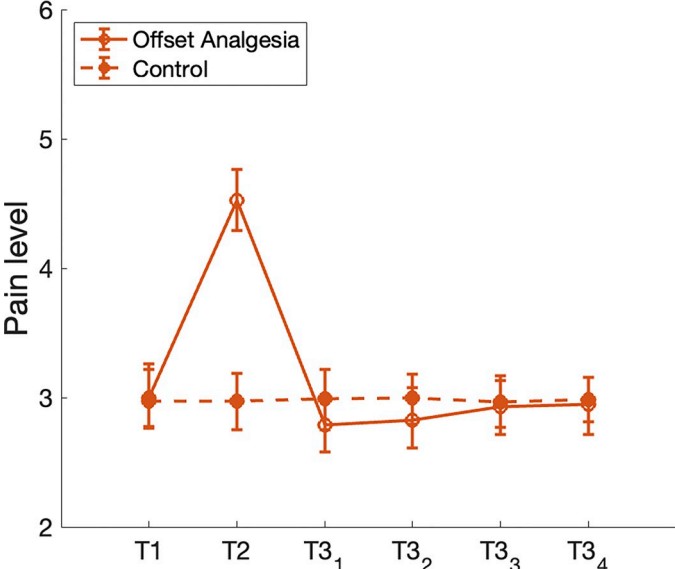

**Fig 7. Results of the hierarchical Kalman filter simulations of offset analgesia.** Offset analgesia is defined as a disproportionally large pain decrease after a minor noxious stimulus intensity reduction. This phenomenon is often elicited by applying noxious thermal stimulation of the same temperature in time intervals T1 and T3, separated by time interval T2 with slightly higher temperature. Our simulation results show a qualitatively similar underdamped response in perceived lever of pain (open red circles) to the minor temperature reduction as is observed in experimental studies of offset analgesia. In the control condition a constant temperature is applied throughout all three time intervals, resulting in a constant level of pain (filled red circles). Error bars indicate the inter quartile ranges.

be reduced and the sensory input can be controlled such that it signals lower levels of tissue injury, this should lead to lower levels of perceived pain. While such sensory restoration is challenging to achieve, there exist some approaches that can provide at least partial sensory restoration, for example by electrical stimulation or surgical "rewiring" of damaged nerves. Our model offers a possible explanation for the working mechanism of treatments for neuropathic pain that involve some form of sensory restoration [36–39].

According to our model, another possible target for relieving neuropathic pain could be the control input, $\boldsymbol{u}$. The more uncertainty there is in the sensory input, and the less uncertainty there is in the prior, the more the perceived level of pain will correspond to the prior. Interestingly, others have shown that uncertainty in the prior may have additional pain modulatory effects, beyond the influencing the weighting of the prior on the perceived level of pain [40]. Thus, providing control input that is certain (low $Q$) to be associated with low levels of pain may be a possible avenue for relieving neuropathic pain. One example of a treatment that may be leveraging this mechanism is mirror therapy, which is commonly used for phantom limb pain [41,42]. Phantom limb pain is a particular form of neuropathic pain, where pain is experienced to arise in a missing limb, for example following amputation. In mirror therapy a mirror is used to create a visual illusion that the missing limb is intact. The working mechanism and efficacy of mirror therapy is disputed [43,44], although it seemingly does provide pain relief to some individuals [42,45,46]. In our model, the visual representation of the limb could be a form of control input. The value of $\hat{B}$ and the associated process noise $Q$ are factors that would determine how the control input influences the perceived level of pain. Probing how control input and its associated uncertainty might influence neuropathic pain could shed light on why mirror therapy, and other similar pain interventions [41,47,48], provide pain relief for some individuals and not for others.

In addition to giving clues of possible avenues for pain prevention and relief, our model also offers a prediction for how different levels of sensory disruption may result in different pain characteristics. Matching sensory symptoms to pain mechanisms is a challenge that researchers and clinicians are grappling with, to better predict treatment responses of individual patients [49,50]. Regarding neuropathic pain, certain sensory characteristics have been found to be particularly common, such as burning pain, numbness, paresthesia (tingling or prickling sensation, "pins and needles"), and pain attacks, sudden bursts of increased pain, or electric shock-like sensations [51–54]. The simulation results presented in Fig 5 indicate that moderately high values or $R$ result in an intermediate level of pain with relatively little variance, which might correspond to a monotonous pain sensation or paresthesia reported by some individuals with neuropathic pain. For higher values of $R$, more diverse pain dynamics are possible. For example, for $\hat{A} \approx 1$ at the time of the nerve injury, large variance of pain intensity is obtained. Under these conditions pain may vary between high and low intensities, possibly reflecting the pain attacks-descriptor that is often used when screening for neuropathic pain. Thus, our model suggests that monotonous or paresthetic sensations of intermediate intensity might be more common in patients with partial denervation ($R$ intermediate) while pain attacks, monotonously high intensity or complete absence of pain would be expected in patients with complete denervation ($R$ high). How these predictions align with clinical findings remains to be determined.

While the hierarchical Kalman filter allows for the internal model parameters to be estimated online, rather than being fit post-hoc from data, it is important to note that other model parameters still must be manually tuned. Parameters that are particularly important for the behavior of the model are the covariances $R$, $Q$ and $Q_p$, as they govern how the state and parameter estimates are updated. $R$ reflects the uncertainty of sensory input and $Q$ reflects the predictability of the pain. The role of $Q_p$ is two-fold: it introduces noise in the parameter estimation, and it governs how much the estimated internal model parameters are updated when there is a prediction error $\hat{e} = \hat{x} - \hat{\hat{x}}$. Consequently, the tuning of $Q_p$ is crucial for the "learning" aspect of the model. Here, we have focused on the role of $R$ and its relevance for neuropathic pain. Similar analyses of the roles of parameters $Q$ and $Q_p$ may provide additional insights into the underlying mechanisms of various pain conditions. As we have varied the value of $R$, we have kept the other parameters as consistent as possible across simulations. This "one size fits all"-approach is likely a contributing factor to why there are quantitative discrepancies between our simulations and experimental data, since the model parameters are likely to vary depending on the context, and possibly also the individual, in which the pain is experienced.

We have focused on modeling pain at the information processing level and have assumed a linear relationship between tissue damage and sensory input arriving to the central nervous system. However, there are a myriad of physical, chemical, and neurobiological processes that modulate sensory afferent signals before they reach the brain. Due to these mechanisms, the mapping from actual or potential tissue damage to sensory input can be quite complex. As an example, thermal discriminative abilities are modified by the temperature to which the skin has been adapted prior to delivering the thermal stimuli, indicating that the relationship between thermal stimulation and sensory input may be non-linear [55]. Omission of this relationship, along with other modulatory mechanisms such as sensitization and habituation, could be an explanation for why our offset analgesia simulations produced an effect that is smaller than typically observed in experiments [56,57] or demonstrated in previous models [58]. This limitation also means that our model may be limited in describing pain phenomena that arise because of impairment of certain pain modulatory mechanisms, as is the case in certain forms of chronic pain.

Another limitation of the current model is the lack of actions and decision making. As a mechanism of protection from harm, pain should elicit actions that allow the organism to withdraw from harmful situations that is cause pain and inform decisions on how to avoid such situations in the future. A few different frameworks have been suggested to model this process, commonly involving some variation of reinforcement learning [59,60]. Active inference is another framework that has been suggested to be relevant for the context of pain [12]. This framework has a lot in common with the model presented here, such as updating a internal model of the world, and also includes the possibility of holding the model fixed and altering actions within the world to sample information that better reflects the predictions [61]. Incorporating the feedback between actions and pain perception in our model could provide insights to how behavior might contribute to persistence of pain and what actions might be helpful for pain relief.

Finally, throughout this work we have entirely focused on the *sensory* aspect of pain. According to the definition, pain is a sensory *and emotional* experience, yet pain is often treated as a purely sensory experience both in clinical practice and in research. Meanwhile, chronic pain is commonly comorbid with psychological disorders such as depression and PTSD [62–64], although the linking mechanism between these conditions remains unclear. Ignoring such a significant portion of the pain experience creates a risk of overlooking mechanisms that might contribute to persistence of pain in chronic pain conditions, as well as possible avenues for new and improved pain treatments. For example, neuroimaging studies have found that initially greater functional connectivity within certain motivation–valuation circuitry predicts pain persistence [65] and that spontaneous fluctuations of chronic pain intensity is associated with activity in brain regions that are typically ascribed to emotional processing [66]. Furthermore, there is tentative evidence that incorporating psychological components in pain treatments yields significant improvements in pain relief [26]. It is possible that existing frameworks, for example within psychological and behavioral science, could hold some of the keys to unlocking the relevance to both decision-making and emotion in the context of pain. We believe that modeling these aspects of pain is a highly interesting avenue for future research, as it could provide insight to previously underexplored mechanisms contributing to pain persistence and relief.

## Methods

### Computational modeling–Single-layer Kalman filter

In modeling the process of pain perception as a Kalman filter, we suggest that the posterior estimate $\hat{x}$ represents the perceived pain, which is an estimate of the true level of actual or potential tissue damage, $x$. Here, potential tissue damage refers to changes in tissue integrity which may not leave lasting damage, such as transiently invoked changes in temperature or brief application of force. For the sake of brevity, we will refer to $x$ simply as *tissue* damage moving forward. The level of tissue damage is influenced by a wide range of overlapping and distinct mechanisms depending on the specific pain that is experienced. Such mechanisms include, but are not limited to, external stimuli being applied to the tissue, healing, or recovery of tissue integrity in the case of potential tissue damage, and physiological processes such as sensitization and inflammation. However, we emphasize that we are *not* attempting to provide a precise model the dynamics of tissue damage. What we aim to do here is to model *how the brain models* tissue damage, and how the resulting predictions from this internal modelling process in the brain may influence the perception of pain. In keeping with Kalman filter terminology, we suggest that the brain's internal model of how tissue damage evolves over time generates a prior estimate, $\hat{\bar{x}}$, which in the context of pain corresponds to the expected level of pain.

At each time point $k$, we assume that there are two main factors contributing to the prior estimate: the level of perceived pain in the previous time step, $\hat{x}^{(k-1)}$, and the control input, $\boldsymbol{u}^{(k)}$, which corresponds to factors that may predict changes in the perceived level of pain, such as motor actions, environmental context and cues, and possibly even psychological processes such as thoughts and emotions. We note that our definition of control input differs somewhat from how control input is typically defined in control theory and engineering applications, where this term refers to an external input or force that is deliberately applied to the system to influence its behavior or state. The traditional definition of control input would be applicable if we were interested in modelling the actual tissue damage, but as we have stated previously that is not the goal of this model. Instead, we are trying to model how the brain models tissue damage. As such, we find that the control input to the brain's internal model of tissue damage instead have to be factors that may predict changes in the perceived level of pain. Furthermore, defining control inputs in this way puts our model in agreement with how priors have been defined in previous Bayesian models of pain [19–22,27].

For ease of communication, we use linear concepts in this model, but the ideas we are exploring extend also to nonlinear concepts. Thus, the prior estimate, or expected level of pain, is obtained as

$$\hat{\bar{x}}^{(k)} = \hat{A}\hat{x}^{(k-1)} + \hat{B}\boldsymbol{u}^{(k)} + \epsilon(0, Q),$$

where $\hat{A}$ and $\hat{B}$ are the parameters of the internal model and $\epsilon(0, Q)$ is the process noise, which reflects the variability not captured by $\hat{A}$ and $\hat{B}$ in the prediction of how pain evolves over time. To reflect that control input can introduce additional uncertainty into the predictions, leading to an increase in process noise compared to when the predictions are evolving without control input, we let the variance of the process noise be $Q = Q_0 + \|\boldsymbol{u}^{(k)}\|_2 Q_u$. However, we do not fully explore the implications of this heteroscedasticity in the examples presented here.

In the context of pain, we suggest that $\hat{A}$ reflects how the pain is expected to develop in absence of control input. If $0 < \hat{A} < 1$, the pain is expected to diminish, and if $\hat{A} > 1$ the pain is expected to worsen with time. The second internal model parameter, $\hat{B}$, reflects how the gain in pain predicted from control input $u$. If $\hat{B} < 0$ pain is expected to decrease, and $\hat{B} > 0$ predicts an increase in pain. The variance, or uncertainty, of the prior estimate is computed as $\bar{P}^{(k)} = \hat{A}^{(k)^2} P^{(k-1)} + Q$, where $P^{(k-1)}$ is the variance of the posterior estimate at time point $k{-}1$.

According to optimal Bayesian integration, the posterior estimate, $\hat{x}$, is a combination of the prior estimate and the sensory input. The sensory input from sensory receptors relating to tissue damage is defined as

$$z^{(k)} = Hx^{(k)} + \epsilon(0, R),$$

where $H$ is the observation matrix mapping from state space (tissue injury) to measurement space (sensory input) and $\epsilon(0, R)$ is the measurement noise, where $R$ denotes the variance in how the sensory input relates to the level tissue damage. For simplicity, we let $H = 1$ and dimensionless. Finally, the posterior estimate, which corresponds to the perceived level of pain, is obtained as

$$\hat{x}^{(k)} = \hat{\bar{x}}^{(k)} + K^{(k)}\left(z^{(k)} - \hat{\bar{x}}^{(k)}\right),$$

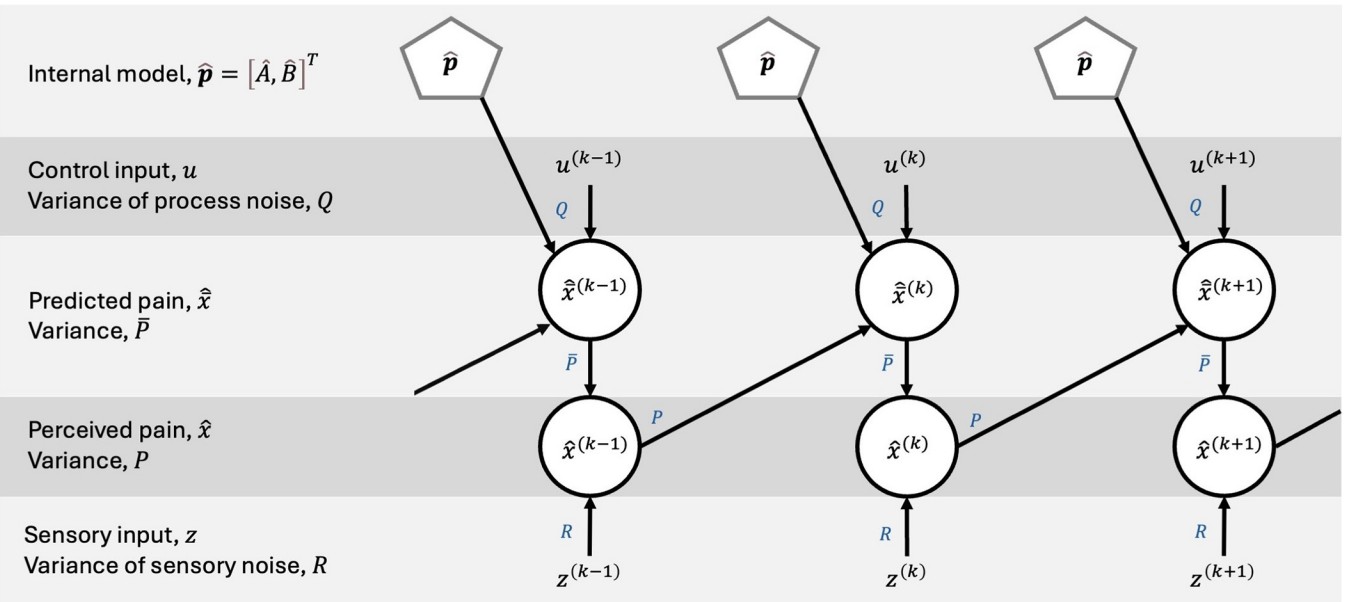

**Fig 8. Schematic figure of the Kalman filter model, depicting the evolution of states across iterations and the relationship between model variables and parameters.** The related variance/noise/covariance to each parameter is indicated next to the arrows. "Top-down" predictions, $\hat{\bar{x}}$, are formed by the control input, $u$, and the internal model, $\hat{\boldsymbol{p}} = [\hat{A}, \hat{B}]^T = [\hat{A}, \hat{b}_1, \ldots, \hat{b}_m]$, where $\hat{b}_i = [\hat{B}]_i$. The prediction is combined with "bottom-up" sensory input, $z$, to form the final estimate, $\hat{x}$, which, in our model, reflects the perceived level of pain.

where $K^{(k)}$ denotes the Kalman gain, which is defined as

$$K^{(k)} = \frac{\bar{P}^{(k)}}{\bar{P}^{(k)} + R}.$$

The posterior estimate has variance $P^{(k)} = (1 - K^{(k)})\bar{P}^{(k)}$, reflecting the uncertainty of the estimate. In Fig 8 we provide a schematic visualization of the model, including the evolution of states across iterations and the relationship between model variables and parameters.

An overview of the variables and parameters in the single-layer Kalman filter is given in Table 1, along with dimensionality and, where applicable, the default value used in simulations when nothing else is specified.

## Computational modeling–Hierarchical Kalman filter

This model builds on previous work on motor adaptation [15–18]. Here, pain is estimated using a Kalman filter, and second Kalman filter is used to update the parameters of the internal model, $\hat{\boldsymbol{p}} = [\hat{A}, \hat{B}]^T = [\hat{A}, \hat{b}_1, \ldots, \hat{b}_m]$, where $\hat{b}_i = [\hat{B}]_i$, based on the difference between the predicted and the perceived pain. This reflects that people integrate previous experiences of pain in their future expectations. The prior estimate of the internal model parameters is given by

$$\hat{\bar{\boldsymbol{p}}}^{(k+1)} = \hat{\boldsymbol{p}}^{(k)} + \epsilon(0, Q_p),$$

where $\epsilon(0, Q_p)$ is noise in the internal model with covariance matrix $Q_p$. The prior estimate of the internal model parameters has covariance $\bar{P}_p^{(k+1)} = P_p^{(k)} + Q_p$, where $P_p^{(k)}$ is the covariance of the internal model estimate at iteration $k$.

For simplicity we assume that $\hat{A}$ and all elements of $\hat{B}$ are independent random variables, and thus the covariance matrix collapses to a diagonal matrix with the variance of the noise of

**Table 1. Overview of the variables and parameters in the single-layer Kalman filter along with dimensionality and, where applicable, the default value used in simulations when nothing else is specified.**

| Parameter | Dimensionality | Default value |
|---|---|---|
| Actual or potential tissue damage, $x$ | 1×1 | \ |
| Perceived pain, $\hat{x}$ | 1×1 | \ |
| Sensory input, $z$ | 1×1 | \ |
| Observation matrix, $H$ | 1×1 | 1 |
| Variance of sensory noise, $R$ | 1×1 | $0.8^2$ |
| Expected pain, $\hat{\bar{x}}$ | 1×1 | \ |
| Control input, $\boldsymbol{u}$ | $m$×1 | $|u| = \{0, 3.7\}$ |
| Estimated pain persistence coefficient, $\hat{A}$ | 1×1 | 0.8 |
| Estimated control gain, $\hat{B}$ | 1×$m$ | $[\hat{B}]_{i=1,\dots,m} = 0.6$ |
| Baseline variance of process noise, $Q_0$ | 1×1 | $0.4^2$ |
| Variance of process noise from control input, $Q_u$ | 1×1 | $0.4^2$ |
| Prior estimate variance, $\bar{P}$ | 1×1 | \ |
| Kalman gain, $K$ | 1×1 | \ |
| Posterior estimate variance, $P$ | 1×1 | \ |
| Initial posterior estimate variance, $P^{(0)}$ | 1×1 | $10^6$ |

each of the elements in the vector of parameters as the diagonal elements, as follows

$$Q_p = \begin{bmatrix} Q_A & \cdots & 0 \\ \vdots & \ddots & \vdots \\ 0 & \cdots & Q_{b_m} \end{bmatrix}$$

To form the posterior estimate of the internal model parameters, the prior estimate is combined with the prediction error $\hat{e}^{(k+1)} = \hat{x}^{(k)} - \hat{\bar{x}}^{(k)}$, which has variance $R_p^{(k)} = P^{(k)} + \bar{P}^{(k)}$. This definition of the prediction error variance differs somewhat from that used in previous published version of the model. Previous definitions include $R_p^{(k)} = R$ [15], $R_p^{(k)} = Q + R$ [16,17] and $R_p^{(k)} = Q + P$ [18]. Our reasoning for updating this definition is simply that we assume that the covariance of the error should be a sum of the covariance of the terms used to compute the error, namely $P^{(k)}$ for $\hat{x}^{(k)}$ and $\bar{P}^{(k)}$ for $\hat{\bar{x}}^{(k)}$. When checking the implications of this update, the model behavior aligns with our expectations. The internal model update takes the form,

$$\hat{\boldsymbol{p}}^{(k+1)} = \hat{\boldsymbol{p}}^{(k)} + K_p^{(k+1)}(\hat{x}^{(k)} - \hat{\bar{x}}^{(k)}) + \epsilon(0, Q_p)$$

$$= \hat{\bar{\boldsymbol{p}}}^{(k+1)} + K_p^{(k+1)}\hat{e}^{(k+1)},$$

where $K_p^{(k+1)}$ is the Kalman gain. Like the traditional Kalman filter, the Kalman gain is defined by the uncertainty of the terms that are combined to form the posterior estimate,

$$K_p^{(k+1)} = \bar{P}_p^{(k+1)} H_p^{(k)} T [H_p^{(k)} \bar{P}_p^{(k+1)} H_p^{(k)^T} + R_p^{(k)}]^{-1},$$

where $H_p$ is the mapping from parameter space to state space, defined as

$$H_p^{(k)} = \frac{\partial \hat{x}}{\partial \hat{\boldsymbol{p}}} = \left[ \frac{\partial \hat{x}^{(k)}}{\partial \hat{A}^{(k)}} \quad \frac{\partial \hat{x}^{(k)}}{\partial \hat{B}^{(k)}} \right] = [H_A \quad H_B] = \left(1 - K^{(k)}\right) \left[ \hat{x}^{(k-1)} \quad \boldsymbol{u}^{(k)} \right].$$

Finally, the covariance of the estimated parameters is given by

$$P_p^{(k+1)} = \bar{P}_p^{(k+1)} \left(I - H_p^{(k)} K^{(k)} K_p^{(k+1)}\right).$$

This definition of $P_p^{(k+1)}$ also differs somewhat from what has been used in previous published versions of the model. The difference here is the introduction of $K^{(k)}$ as a scaling factor to $H_p^{(k)} K_p^{(k+1)}$. Without the scaling factor $K^{(k)}$ the covariance of the estimated parameters, $P_p^{(k+1)}$, will tend to 0 when $R \rightarrow \infty$. When there is infinite sensory noise there is no reliable input to inform the accuracy of the estimated internal model parameters, and thus $P_p^{(k+1)}$ should not tend to 0 (since $P_p^{(k+1)} \approx 0$ indicates high confidence in the estimated parameters). To address this contradictory behavior, we introduce the scaling factor $K^{(k)}$.

Estimation of the level of pain, $\hat{x}$, is similar to the process outlined for the single-layer Kalman filter, with a few key changes relating to the prior estimate First, the internal model parameters change over time, requiring us to specify the temporal index of the parameters. Second, the term $H_p^{(k)} P_p^{(k)} H_p^{(k)^T}$ has been added to the calculation of the prior estimate variance, $\bar{P}^{(k)}$, to reflect that the uncertainty in the internal model estimate influences the uncertainty in the state estimate. We note that, in the single-layer Kalman filter, the baseline process noise, $Q_0$, has to include this source of uncertainty in addition to other factors that may cause variability in the prior estimate. Since we can explicitly account for the uncertainty here introduced by $P_p$ in the prior estimate, by the addition of the term $H_p^{(k)} P_p^{(k)} H_p^{(k)^T}$ in the prior estimate variance, we update the value of the baseline process noise to be $Q_0 = 0.1^2$ (as compared to $Q_0 = 0.4^2$ in the single-layer Kalman filter). Taken together, these changes imply that the prior estimate is computed as

$$\hat{\bar{x}}^{(k)} = \hat{A}^{(k)} \hat{x}^{(k-1)} + \hat{B}^{(k)} \boldsymbol{u}^{(k)} + \epsilon(0, Q),$$

where $Q = Q_0 + \|\boldsymbol{u}^{(k)}\|_2 Q_u$ just as for the single-layer Kalman filter, but with the updated value of $Q_0$. The variance of the prior estimate is $\bar{P}^{(k)} = \hat{A}^{(k)^2} P^{(k-1)} + Q + H_p^{(k)} P_p^{(k)} H_p^{(k)^T}$.

In Fig 9 we provide a schematic visualization of the hierarchical Kalman filter model, including the evolution of states and parameters across iterations and the relationship between model variables and parameters.

An overview of the variables and parameters in the hierarchical Kalman filter is given in Table 2, along with dimensionality and, where applicable, the default value used in simulations when nothing else is specified.

## Simulations

Since pain frequently is measured on an 11-point scale, where 0 = no pain and 10 = worst pain imaginable, we let tissue damage $x$ exist on a similar 11-point scale of arbitrary units. For simplicity, we have set $H = 1$ and dimensionless such that sensory measurements $z$ also exist on an 11-point scale of arbitrary units. For each time step $k$ we draw sensory input $z^{(k)} \sim N(x^{(k)}, R)$, truncated at 0 and 10. Truncation is achieved by resampling if the drawn sample falls outside of the permitted range. To account for asymmetry in the resulting distributions due to the truncation, we have chosen to report medians and interquartile ranges, rather than means and

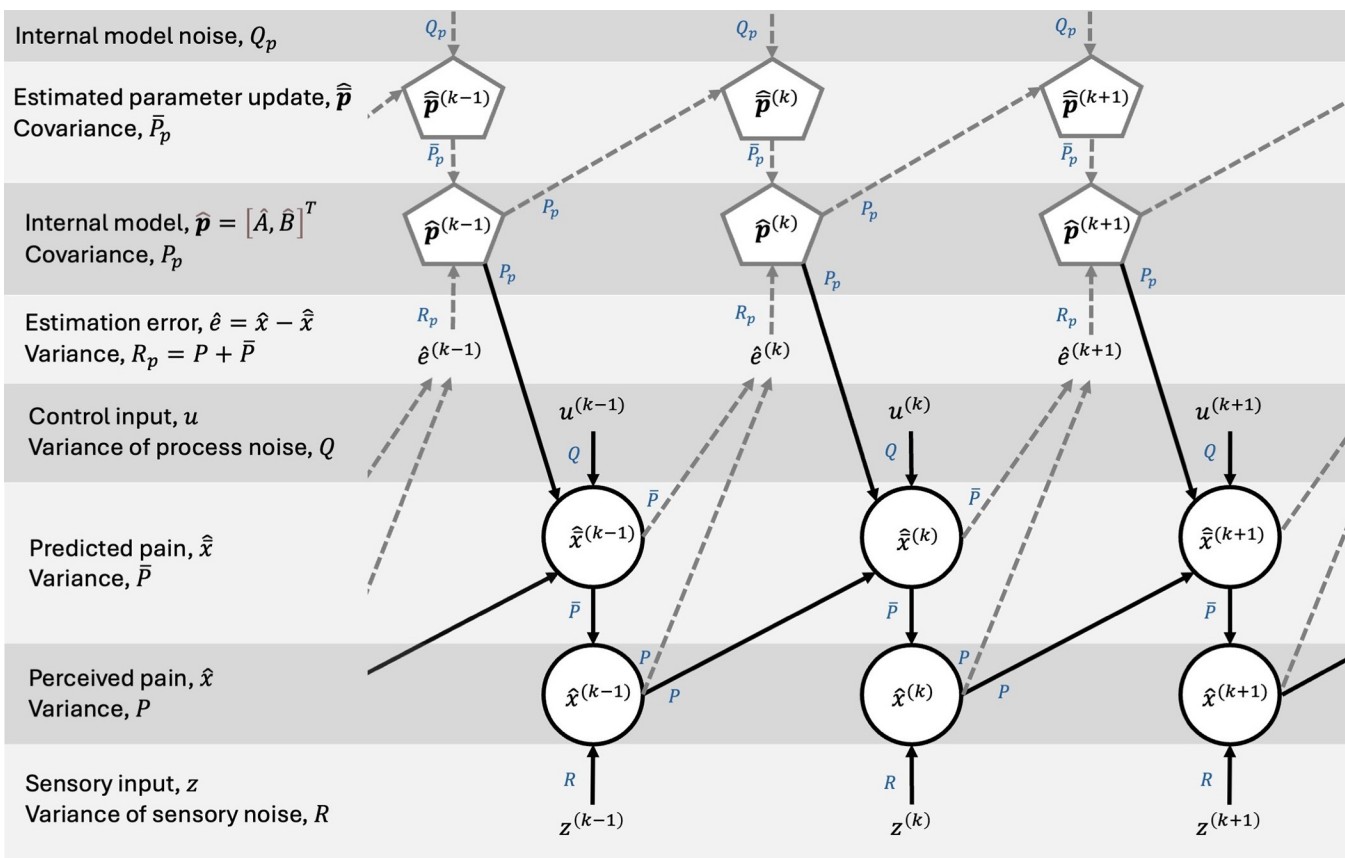

**Fig 9. Schematic figure of the hierarchical Kalman filter model, depicting the evolution of states and internal model parameters across iterations, and the relationship between model variables and parameters.** The related variance/noise/covariance to each parameter is indicated next to the arrows. State estimation (lower part of the figure with arrows in solid black lines) is comparable to the single-layer Kalman filter model depicted in Fig 8. The estimation error, i.e., the difference between the predicted and perceived pain, $\hat{e} = \hat{x} - \hat{\bar{x}}$, acts as the "bottom-up" input to the parameter estimation Kalman filter (upper part of the figure with arrows in dashed grey lines).The estimated parameters from the previous iteration (and additive internal model noise, $Q_p$) act as the prior estimate, $\hat{\bar{p}}$. Finally, the prior estimate, $\hat{\bar{p}}$, and the estimation error, $\hat{e}$, are combined to form the new estimated internal model parameters.

standard deviations. These values are computed across 100 simulations, unless otherwise specified.

Matlab code for all simulation results presented in this article are available online at: https://zenodo.org/doi/10.5281/zenodo.10960405.

**Fig 1B, 1E and 1F—Placebo hyperalgesia and nocebo hypoalgesia.** For placebo hypoalgesia and nocebo hyperalgesia we set our simulations up to resemble with the experimental paradigm of described by Jepma et al. [20], to allow for comparison between our simulation results and the experimental data that is available open source at https://osf.io/bqkz3/. Here, we focus on Study 1, which consisted of a learning phase and a test phase (similar results for Study 2 can be found in Fig A in S1 Supplementary Results). On each trial in the learning phase a visual cue (a geometric figure) was presented to the participants. Following the cue, participants indicated which heat level they expected on a 100-unit visual analogue scale (VAS). After rating their expected pain, a picture of a thermometer appeared indicating low heat level for low-pain cues, and high heat levels for high-pain cues. In the testing phase each trial would consist of one of the cues from the preceding learning phase being presented, followed by the participants rating their expected level of pain on the 100-unit VAS. After the

**Table 2. Overview of the variables and parameters in the hierarchical Kalman filter along with dimensionality and, where applicable, the default value used in simulations when nothing else is specified.** Note that $I_{(1+m)}$ denotes a $(1+m)$-dimensional identity matrix.

| Parameter | Dimensionality | Default value |
|---|---|---|
| Actual or potential tissue damage, $x$ | $1 \times 1$ | \ |
| Perceived pain, $\hat{x}$ | $1 \times 1$ | \ |
| Sensory input, $z$ | $1 \times 1$ | \ |
| Observation matrix, $H$ | $1 \times 1$ | 1 |
| Variance of sensory noise, $R$ | $1 \times 1$ | $0.8^2$ |
| Expected pain, $\bar{\hat{x}}$ | $1 \times 1$ | \ |
| Control input, $\boldsymbol{u}$ | $m \times 1$ | $\|u\| = \{0, 3.7\}$ |
| Estimated internal model parameter, $\hat{A}$ | $1 \times 1$ | \ |
| Initial value of $\hat{A}$, $\hat{A}^{(0)}$ | $1 \times 1$ | 0.8 |
| Estimated internal model parameter, $\hat{B}$ | $1 \times m$ | \ |
| Initial value of $\hat{B}$, $\hat{B}^{(0)}$ | $1 \times m$ | $[\hat{B}^{(0)}]_{i=1,\ldots,m} = 0.6$ |
| Alternative notation for estimated internal model parameters, $\hat{\boldsymbol{p}}$ | $(1+m) \times 1$ | \ |
| Baseline variance of process noise, $Q$ | $1 \times 1$ | $0.1^2$ |
| Variance of process noise from control input, $Q_u$ | $1 \times 1$ | $0.4^2$ |
| Prior estimate variance, $\bar{P}$ | $1 \times 1$ | \ |
| Kalman gain for pain estimation, $K$ | $1 \times 1$ | \ |
| Posterior estimate variance, $P$ | $1 \times 1$ | \ |
| Initial posterior estimate variance, $P^{(0)}$ | $1 \times 1$ | $10^6$ |
| Mapping from parameter space to state space, $H_p$ | $1 \times (1+m)$ | \ |
| Covariance of internal model noise, $Q_p$ | $(1+m) \times (1+m)$ | $0.002^2 I_{(1+m)}$ |
| Prior parameter estimation, $\bar{\hat{\boldsymbol{p}}}$ | $(1+m) \times 1$ | \ |
| Covariance of prior parameter estimation, $\bar{P}_p$ | $(1+m) \times (1+m)$ | \ |
| Kalman gain for parameter estimation, $K_p$ | $(1+m) \times (1+m)$ | \ |
| Covariance of parameter estimation, $P_p$ | $(1+m) \times (1+m)$ | \ |
| Initial covariance of parameter estimation, $P_p^{(0)}$ | $(1+m) \times (1+m)$ | $10^6 I_{(1+m)}$ |
| Pain prediction error, $\hat{e}$ | $1 \times 1$ | \ |

expected pain rating, a noxious heat (47˚C or 48˚C, unrelated to the preceding cue) was applied to the participants' left inner forearm. After the thermal stimulation the participants again used the 100-unit VAS to rate their experienced pain. The learning and test phases consisted of 120 and 40 trials, respectively.

With the single-layer Kalman filter, we simulate the 40 test trials as discrete events, were the cue and noxious stimulation on each trial are independent from the cue and stimulation of previous trials. This means that the pain in trial $k$ is independent of the pain in trials $k-1$, i.e., the value of the internal model parameter $\hat{A}$ is $\hat{A} = 0$. Thus, the prior estimate, or expected pain, in these simulations is

$$\bar{\hat{x}}^{(k)} = \hat{B}^{(k)} \boldsymbol{u}^{(k)} + \epsilon(0, Q),$$

where we let the visual cue correspond to the control input, $\boldsymbol{u}^{(k)}$. To distinguish between the placebo and nocebo cues, we let $\boldsymbol{u}^{(k)}$ be a two-dimensional vector such that $\boldsymbol{u}^{(k)}_{placebo} = [u\ 0]^T$ and $\boldsymbol{u}^{(k)}_{nocebo} = [0\ u]^T$ are orthogonal to each other. We let the control input have identical magnitude, $u = 3.7$, in placebo and nocebo trials, to reflect that the cues do not inherently signal different levels of pain. The internal model parameter $\hat{B}$ takes the form $\hat{B} = [\hat{b}_{placebo}, \hat{b}_{nocebo}]$. For

each individual, we draw $\hat{b}_{placebo} \sim N(0.7, 0.5^2)$ and $\hat{b}_{nocebo} \sim N(1.3, 0.5^2)$ (truncated at 0 and $\infty$), to reflect the inter-individual differences in pain expectations. We let the tissue damage elicited by the thermal stimulation be $x_{low}$ = 3.1 for low heat trials (47˚C) and $x_{high}$ = 4.3 for high heat trials (48˚C). We simulate 28 individuals, to match the number of individuals in the experimental data and report the averages and standard errors of the expected and perceived pain in Fig 1.

**Fig 2 –Chronic pain.** In Fig 2, we aim to demonstrate that, under certain circumstances, the perceived pain may be disproportionately high compared to the true level of tissue damage and that the pain may persist even when the tissue damage has recovered. To this end, we simulate tissue damage as $x^{(k)} = Ax^{(k-1)} + B\tilde{u}^{(k)}$, where $x^{(0)}$ = 0, $A$ = 0.9 reflects the rate of recovery, $B$ = 0.8 reflects the gain in tissue damage from $\tilde{u}^{(k)}$, and $\tilde{u}^{(k)}$ corresponds to input that influences the level of tissue damage, such as force or heat applied to the tissue. Sometimes there is a strong correlation between $\tilde{u}^{(k)}$ and $u^{(k)}$, the control input to the prediction stage of the Kalman filter. Examples of such scenarios are if the tissue damage is caused by a motor action, or if there are visual cues predictive of the imminent tissue damage (e.g., seeing a hammer approach your thumb). We simulate this type of scenario in Fig 2 by letting $\tilde{u}^{(k)} = u^{(k)} = \{3.7 \text{ for } k = k^*, 0 \text{ otherwise}\}$. Other times $\tilde{u}^{(k)}$ and $u^{(k)}$ may be completely independent, such as unknowingly being stung by a bee. In this scenario, if you do not see or hear the bee approaching, there are no predictive cues ($u^{(k)}$ = 0), but the bee sting still ($\tilde{u}^{(k)}$) elicits tissue damage which typically also results in pain. We provide simulation results for such a scenario, where $\tilde{u}^{(k)} = \{3.7 \text{ for } k = k^*, 0 \text{ otherwise}\}$ while $u^{(k)} = 0 \,\forall\, k$, in Fig C in S1 Supplementary Results. Default parameter values specified in Table 1 are used in these simulations, except for $\hat{A} = 1$, and the value of $R$ which is $0.8^2$ in the left panel and $1.8^2$ in the right panel.

**Fig 3 –Neuropathic pain.** In Fig 3, different levels of sensory disruption are simulated by letting the value of $R$ vary, and for each value of $R$ the resulting pain is assessed for different values of $\hat{A}$. To reflect the transition to a state of nerve injury more accurately, all the simulation results shown in the figure are preceded by a simulated period of no sensory disruption ($R = 0.8^2$, default) to allow the estimated state $\hat{x}$ and associated uncertainty $P$ to settle into baseline values.

**Fig 4 - Classical conditioning.** The classical conditioning simulations follow similar principles as placebo and nocebo simulations. We, again, we set our simulations up to resemble with the experimental paradigm of described by Jepma et al. [20], and birefly outlined in the description of simulations for *Fig 1B), 1E) and 1F)* above, to allow for comparison between our simulation results and the experimental data. Just as for the single-layer Kalman filter, we let the visual cue correspond to the control input, $\boldsymbol{u}^{(k)}$. To distinguish between the placebo and nocebo cues, we let $\boldsymbol{u}^{(k)}$ be a two-dimensional vector such that $\boldsymbol{u}^{(k)}_{placebo} = [u\ 0]^T$ and $\boldsymbol{u}^{(k)}_{nocebo} = [0\ u]^T$ are orthogonal to each other. We let the control input have identical magnitude, $u$ = 3.7, in placebo and nocebo trials, to reflect that the cues do not inherently signal different levels of pain, and we let the internal model parameter $\hat{B}$ take the form $\hat{B} = [\hat{b}_{placebo}, \hat{b}_{nocebo}]$.

In the single-layer Kalman filter we manually tuned the values of $\hat{b}_{placebo}$ and $\hat{b}_{nocebo}$ to obtain the desired model output. Here, we instead simulate the learning phase of the experiment to let the internal model parameters of the hierarchical Kalman filter be set by the conditioning procedure. Before the onset of the simulated conditioning procedure, the initial values of $\hat{B}$ are drawn from the normal distribution $N(0.9, 0.8^2)$ truncated at 0 and $\infty$. This starting point is chosen based on the reported expectations of pain upon presentation of a new, neutral cue during the test phase of Study 2 in Jepma et al. (see Figs A and B in S1 Supplementary Results

and [20]. Then, across 120 trials, we pair the $\boldsymbol{u}_{placebo}^{(k)} = [u\ 0]^T$ with a low heat stimulation resulting in sensory input $z_{low} \sim N(2.2, R)$, and $\boldsymbol{u}_{nocebo}^{(k)} = [0\ u]^T$ is paired with a high heat stimulation resulting in sensory input $z_{high} \sim N(5.2, R)$. We note that this paradigm differs somewhat from the experimental paradigm in Jepma et al. [20], where the visual cues were paired with a picture of a thermometer indicating low or high heat levels, not with actual thermal stimulation. Placebo and nocebo can be elicited through many different forms of conditioning; by pairing the conditioned stimuli with noxious stimulation, by 'symbolic conditioning' as with the pictures of the thermometers, by verbal suggestion alone, etc. [29]. While the exact neural processes involved in the establishing the cue-pain associations for different forms of conditioning may differ, the resulting behavioral outcomes are qualitatively similar [19–22,29].

We simulate the conditioning procedure for 28 individuals, to match the number of individuals in the experimental data. We then simulate 40 test trials for each individual, with cue-heat pairings matching the experimental data, and report the averages and standard errors of the expected and perceived pain during the test trials in Fig 4. We also visualize how the internal model parameters and expected pain associated with the cues evolve across the conditioning procedure.

**Fig 5 –Spontaneous neuropathic pain.** Similar to Fig 3, we simulate different levels of sensory disruption by varying the value of $R$. A key difference here is that, here, we assess the resulting pain for different *initial values* of $\hat{A}$ at the time of nerve injury. However, due to the adaptive nature of the hierarchical Kalman filter, the value of $\hat{A}$ does not stay constant over time. This also means that we cannot run a baseline period before nerve injury, as in Fig 3, since this would alter the value of $\hat{A}$. Instead, we manually tune the uncertainty of the state and parameter estimations to approximately correspond to the values obtained from other baseline simulations, such that $P = 1$ and $P_p = I$ at the time of nerve injury.

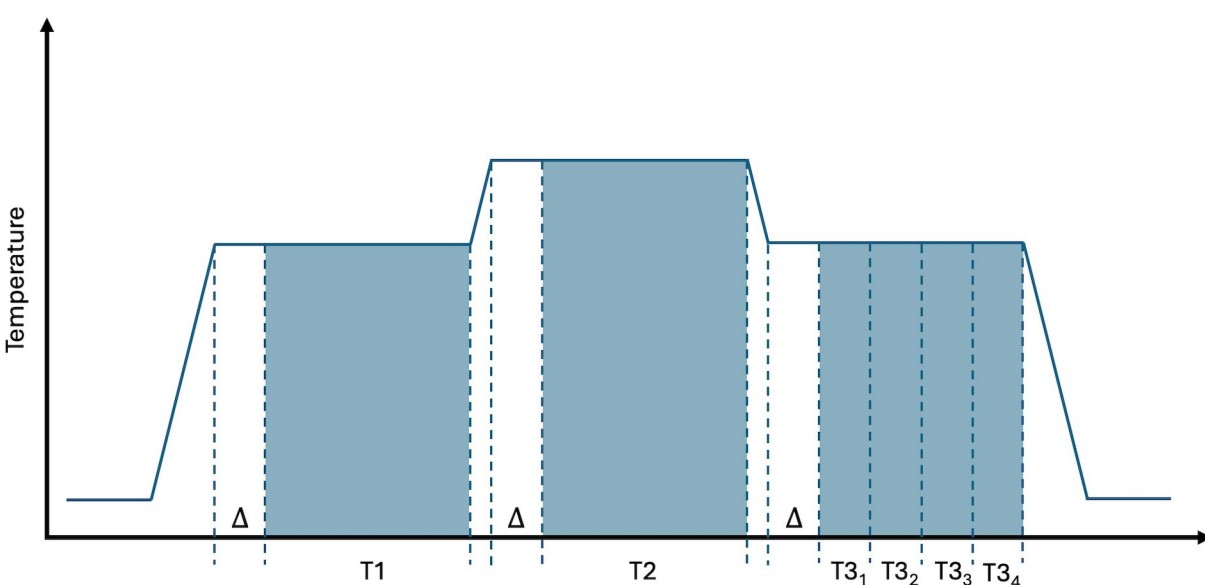

**Fig 10. Pain is assessed in the intervals $\{T1, T2, T3_1, T3_2, T3_3, T3_4\}$ in the offset analgesia simulations.** The first period following ramp-up or ramp-down of temperature, indicated by $\Delta$ in the figure, is omitted to account for any delay in change of perceived pain relative to the change in temperature.

**Fig 6** –**Risk factors for developing neuropathic pain.**   In Fig 6, we aim to demonstrate how historical experiences of pain may influence the risk of developing neuropathic pain following a nerve injury. To this end, we first simulate a period preceding the nerve injury with the baseline values specified in Table 2. Similar to Fig 2, we simulate tissue damage as $x^{(k)} = Ax^{(k-1)} + B\tilde{u}^{(k)}$, where $B = 0.8$, $x^{(0)} = 0$ and $\tilde{u}^{(k)} = u^{(k)} = \{3.7 \text{ for } k = k^*, 0 \text{ otherwise}\}$ with time points $k^*$ indicated by green stars on the x-axes of the figures. The value of $A$ is specified in the title of each subplot. Nerve injury is simulated by increasing the value of $R$ to $800^2$, indicated by a vertical dashed black line in the graphs.

**Fig 7** - **Offset analgesia.**   In experimental paradigms of offset analgesia, the subject typically receives noxious stimuli of varying intensity (temperature) without any cues signaling the upcoming changes in intensity. Therefore, throughout all offset analgesia simulations we let $u = 0$. The level of tissue damage is assumed to be linearly proportionate to the applied temperature. At simulation onset the internal model parameters are set to match the real-world parameters, $\hat{A} = A$ and $\hat{B} = B$. The level of perceived pain is assessed as the median value of $\hat{x}$ in intervals $\{T1, T2, T3_1, T3_2, T3_3, T3_4\}$ as specified in Fig 10. The period immediately following ramp-up or ramp-down of temperature (indicated by $\Delta$ in Fig 10) is omitted to account for any delay in change of perceived pain relative to the change in temperature.

## Supporting information

**S1 Supplementary Results. Description and resulting figures of additional simulations not included in the main manuscript.**
(DOCX)

## Author Contributions

**Conceptualization:** Malin Ramne, Jon Sensinger.

**Formal analysis:** Malin Ramne.

**Software:** Malin Ramne.

**Supervision:** Jon Sensinger.

**Writing – original draft:** Malin Ramne.

**Writing – review & editing:** Jon Sensinger.

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
