## [Decision Letter · Decision Letter 0]

16 Jul 2024

Dear Mx Ramne,

Thank you very much for submitting your manuscript "A Computational Framework for Understanding the Impact of Prior Experiences on Pain Perception and Neuropathic Pain" for consideration at PLOS Computational Biology.

As with all papers reviewed by the journal, your manuscript was reviewed by members of the editorial board and by several independent reviewers. In light of the reviews (below this email), we would like to invite the resubmission of a significantly-revised version that takes into account the reviewers' comments.

The guest editor and reviewers acknowledge the novelty and importance of providing a theoretical framework to describe pain perception and its modulation. The reviewers have raised a number of major and minor points that would warrant a major revision. The main areas of concern are:

1) Clarity, especially regarding the explanation of modelling choices and results

2) Modelling - The simulation results could be validated by fitting existing, open datasets (see Reviewer 1 comments)

3) The conceptual definition of a number of terms (see Reviewer 2 comments)

We cannot make any decision about publication until we have seen the revised manuscript and your response to the reviewers' comments. Your revised manuscript is also likely to be sent to reviewers for further evaluation.

Sincerely,

Flavia Mancini

Guest Editor

PLOS Computational Biology

Thomas Serre

Section Editor

PLOS Computational Biology

The guest editor and reviewers acknowledge the novelty and importance of providing a theoretical framework to describe pain perception and its modulation. The reviewers have raised a number of major and minor points that would warrant a major revision. The main areas of concern are:

1) Clarity, especially regarding the explanation of modelling choices and results

2) Modelling - The simulation results could be validated by fitting existing, open datasets (see Reviewer 1 comments)

3) The conceptual definition of a number of terms (see Reviewer 2 comments)

Reviewer's Responses to Questions

**Comments to the Authors:**

Reviewer #1: Uploaded as attachment.

Reviewer #2: In the manuscript titled "A Computational Framework for Understanding the Impact of Prior Experiences on Pain Perception and Neuropathic Pain", authors Ramne and Sensinger expand a Bayesian approach (single-layer Kalman filter) of modelling pain genesis to also include parameter updates over time, using a second layer for hierarchical Kalman filtering. This allows for consideration of _concurrent processes_ like expectation, as well as the _formation_ of the respective process parameters via previous experiences. Their models (partially) replicate/mimick several known physiological & pathological pain phenomena, including expectancy-based & conditioning-based placebo/nocebo effects, offset analgesia, chronic pain & neuropathic pain arising from increased sensory uncertainty. The article is well-written and the methods are intriguing, and the approach could provide an overall useful tool (for computational modelling, obviously, but also for hypothesis generation and detecting commonalities and differences between pain entities). However, I would contest some imprecisions and limitations of the models, as well as point out general shortcomings of the manuscript. Many of those points are not "this is wrong" but rather "please consider".

I. MAJOR ISSUES

[for the first two comments I have no good suggestion on how to deal with them but I encourage the authors to work on these parts; if there was a more coherent characterization of the parameters of the models, readability for a wider audience might improve]

I.1. The authors consider both "actual and potential tissue damage" for noxious input, as per IASP definition. I have a semantic and a conceptual concern about the current framing: Semantically, the current characterization of parameter A (or ^A, respectively) is a reference to "healing", but if only potential tissue damage takes place (say, transient experimental pain without actual injury), what does "healing" refer to? Maybe "recovery of baseline function" or some such formula is a more suitable description, as it could pertain to compensation or recovery (recovered tissue integrity, or recovered sensitivity, or even loss of memory of the painful episode). Conceptually, the authors do point out that the level of abstraction of the models is rather high, i.e. does not posit any particular implementation or mechanism, just describes the outcome of whatever mechanism might be at work. Ultimately, I would like them to point out that there are overlapping and distinct mechanisms at work depending on the specific pain that is experienced.

I.2. Relatedly, from a psychological perspective, I am not satisfied with the descriptions for processes at work during time points k and k-1, namely "actions" Bu and "healing" A (see above). At its core, Bu(k) and Ax(k-1) agnostically refer to _all processes pertaining to the respective time points k or k-1_ (however weighted). Colloquially and in some psychological settings, "action" would refer to overt and deliberate actions; while it is fine (and possibly conventional) to use the word in this case, I would at least clarify that it can pertain to any processes (e.g. neuropsychological processes like attention/distraction, physiological processes, motor behaviors; also, although the authors later suggest this is not the case, descending modulation). This is hinted at in the methods section but would clarify things if it was mentioned earlier.

I.3 Line 244. "First, we show that there is no expectation of pain associated with the conditioned stimulus at baseline." I was having trouble grasping how this was meant. In the design implicit in the sequence of events of Figure 4, the results would indicate an _expectation of no/zero pain_ (from the zero pain applied at T1/Baseline, which "pulls down" the expectation initially), not _no expectation_. In practice, this is not how the CS is introduced as this would probably lead to latent inhibition. Instead, the CS being neutral indicates that is it conditionable, not that it is conditioned to zero pain. Put bluntly, in placebo conditioning, the CS is not shown with the baseline stimulation (besides, "zero pain" is not a baseline condition. I am not sure how to amend this model-wise, but I feel the placebo/nocebo conditioning part is unconvincing as is.

I.4 Line 249ff. From what I see, the nocebo results do _not_ match the single-layer Kalman results in Fig 1, not even "qualitatively". This may be related to deficiencies in the model pointed out in I.3; plot inspection indicates that the conditioning procedure never overcame the "zero pain" conditioning from the baseline but still produced a placebo effect despite (ostensibly) nocebo conditioning. The statement that there is replication of placebo/nocebo conditioning seems unwarranted at this stage.

II. MINOR ISSUES (semantic & conceptual)

- The first paragraph of the Results have an Introductory feel to them.

- Line 133 "intensity, quality, and characteristics" - are intensity and quality not characteristics? Maybe an "other" is missing here...

- Line 135ff. At first reading, I found the elaboration of the formulae in the Results section a bit lacking (possibly owing to the imho unfortunate PLOS-sided postponing of the Methods). k for example is not defined.

- Figure 2. "True tissue damage" may be more suitably described as "nociceptive signal" or "afferent drive" (see I.1)?

- Figure 2. Your legend entries were flipped for the second row - solid line is perceived pain, dotted line is "true tissue damage" (or whatever alternative). Incidentally, MATLAB warns that you provide extra legend entries (using the uploaded code).

- Line 183, "In the context of Bayesian models, elevated uncertainty of sensory input is a requirement for expectations to have a significant enough influence that they cause the level of pain to deviate from what is signaled by the sensory input." Very vague statement, why does uncertainty have to be "elevated" and what is "significant enough" or "deviation"? Maybe rephrase to point to the fact that _the more_ likelihood uncertainty and _the less_ prior uncertainty, the more the posterior will correspond to the prior, or so? Anyway please put more concisely.

- Line 193, "The gradual increase of pain before the onset of tissue damage in the bottom right panel of Figure 2" - does this refer to the ~0.1 point increase visible in the graph? I'm not convinced that would constitute "how pain can arise spontaneously" (given the dispersion that makes any increase in that range rather cosmetic), and I don't think your message would be impacted if you did not provide this Figure 2 observation to introduce Figure 3.

- Figure 3. I would find this figure much more informative if you used Rs = [1 10 100]*sKFParams.R; instead of Rs = [1 100 1e6]*sKFParams.R; That way, you could demonstrate that the model does not simply produce ceiling effects, and I don't recall reading that you discussed the initial "overshoot" in the R=600² condition anyway. I appreciate though that for Figure 5 you want to preserve R=600², but four panels also wouldn't hurt as the individual panels are not overloaded with information anyway.

- Line 236ff. I would add that the conditioned response in placebo/nocebo conditioning would be the processes accompanying the painful stimulus, presumably "the state anticipating a noxious stimulus of intensity x" (possibly even "psychomotoric readiness to give rating in a certain range of the scale", but that might be a bit iconoclastic).

- Discussion could be more conscise, maybe skip the detour to predictive coding (I realize that this paragraph may have been desired by a different reviewer and I'm okay with it, just making the observation that the discussion feels a tad too long).

- Line 364 "To our awareness, this phenomenon has not previously been reproduced in mathematical models of pain." - it has (Cecchi et al, PLOS Comput Biol 2012), and I do recommend checking out their model (using derivatives and many more parameters) as it provides very good approximation to OA (and transient/phasic pain stimulation in general).

- Line 448 "This limitation also means that our model cannot account for pain phenomenon that may arises because of impairment of pain modulation, such as certain forms of chronic pain." Wouldn't Bu(k) account for descending modulation? (see I.2)

- Line 471. I would contest that allodynia describes a behavioral outcome (non-noxious intensity causing pain), not the genesis of that outcome. I.e. allodynia can arise through multiple processes (sensitiziation, inflammation, etc.) and is not solely a neuropathic phenomenon. Please revise.

- Line 518ff. Hm I'm not that sold on prospect theory, the paragraph also seems to conflate appetitive and aversive motivation in regards to pain, which follow different functions (loss = usually losing something good, not = gaining something bad)... is it possible to find a more concrete/analogous application in the aversive domain?

- Line 660. Replace "commonly" with "frequently".

III. MINOR ISSUES (formal)

- Figure 2. Your legend here says "True tissue damage" but all other figures use "Tissue damage".

- Line 423. "electric shocks" => "electric-shock-like sensations"

- Line 423. "The simulation results presented in Figure X indicate that moderately high values or R"... (Figure X => Figure ?, or => of)

- Line 448 "pain phenomenon that may arises"

- Line 481 "it seemingly does to provide pain relief"

- Line 532 "concepts form prospect theory"

- Line 661 "exist on similar 11-point scale"

- References: Some of the references have been mixed up (16==25, 26==49)

IV. CODE ISSUES

- Possibly a MATLAB version issue, but mine throws an error at PlotFigure6:116

% scatter(iEvent, .5, 200, 'filled', 'pentagram', 'MarkerFaceColor',[0.4667 0.6745 0.1882]);

% this should probably be

scatter(iEvent, repmat(.5,size(iEvent)), 200, 'filled', 'pentagram', 'MarkerFaceColor',[0.4667 0.6745 0.1882]);

Reviewer #3: Well made manuscript. Recommend correcting minor grammatical mistakes.

**Have the authors made all data and (if applicable) computational code underlying the findings in their manuscript fully available?**

Reviewer #1: Yes

Reviewer #2: Yes

Reviewer #3: Yes

PLOS authors have the option to publish the peer review history of their article (what does this mean?). If published, this will include your full peer review and any attached files.

Reviewer #1: No

Reviewer #2: **Yes: **Bjoern Horing

Reviewer #3: No
---

## [Decision Letter · Decision Letter 1]

17 Oct 2024

Dear Mx Ramne,

We are pleased to inform you that your manuscript 'A Computational Framework for Understanding the Impact of Prior Experiences on Pain Perception and Neuropathic Pain' has been provisionally accepted for publication in PLOS Computational Biology.

Best regards,

Flavia Mancini

Guest Editor

PLOS Computational Biology

Thomas Serre

Section Editor

PLOS Computational Biology

This study explores how pain perception is influenced not only by sensory input but also by cognitive factors like prior expectations. Using a Bayesian framework, the research demonstrates how the integration of prior expectations and sensory input can explain pain phenomena such as placebo hypoalgesia, nocebo hyperalgesia, chronic pain, and spontaneous neuropathic pain. The findings suggest that the value of the prior, shaped by internal model parameters, plays a critical role in these pain experiences. The study further applies a hierarchical Bayesian model to show how prior experiences, through classical conditioning, contribute to placebo and nocebo effects. Additionally, it explains the phenomenon of offset analgesia, where a small reduction in pain stimulus results in a disproportionately large pain decrease. Simulations of neuropathic pain suggest that persistent non-neuropathic pain or the complete absence of painful experiences could both be risk factors for developing neuropathic pain after denervation. Overall, the results provide valuable insights into how prior experiences shape pain perception and may inform future strategies for pain prevention and management. I recommend accepting this study due to its robust application of Bayesian models and its potential for advancing pain relief strategies.

Reviewer's Responses to Questions

**Comments to the Authors:**

Reviewer #1: Dear Editors, Authors and Readers,

Authors highlight an open, and highly important question in the field of pain neuroscience as to what mechanisms/processes gives rise to prior beliefs, which in turn affect the experienced level of pain. Authors propose the use of a (modified) Kalman Filter (KF) model (an algorithm used to estimate latent states in a specific setup of probabilistic / Bayesian inference) to describe phenomena such as placebo hypoalgesia, nocebo hyperalgesia, offset analgesia and neuropathic pain. The authors succeed in reproducing the stated phenomena (placebo/nocebo, neuropathic pain) through simulation. The proposed application of Hierarchical Kalman Filter in the context of pain is novel and holds promise for modelling pain inference. Additionally, it allows for some of the system parameters to be estimated online “by the participant’s sensory/cognitive system” rather than inferred post-hoc from the data, and as such is a valuable approach. They provide additional support to their simulation results and the novel modelling approach by showcasing key effects alongside a preexisting dataset. The paper tells a well-written, and clearly presented story as to how different parameter settings can be mapped onto biological conditions and past experiences that then contribute to maladaptive pain states. I found the description of the evolution of B parameters of the HKF model giving rise to placebo/nocebo effects convincing and well presented. Additionally, the discussion about potential treatment avenues as informed by the modelling interpretation sounds particularly interesting.

Overall, I’m satisfied with how the authors addressed my previously raised points.

Minor points:

1. (2nd paragraph of Methods Single KF)

“that factors” doesn’t fit here:

“As such, we find that the control input to the brain’s internal model of tissue damage instead have to be factors that factors that may predict changes in the perceived level of pain.”

Reviewer #2: Thank you for the thorough revision, all my issues have been addressed.

**Have the authors made all data and (if applicable) computational code underlying the findings in their manuscript fully available?**

Reviewer #1: Yes

Reviewer #2: Yes

PLOS authors have the option to publish the peer review history of their article (what does this mean?). If published, this will include your full peer review and any attached files.

Reviewer #1: No

Reviewer #2: **Yes: **Bjoern Horing

---

## [Editor Report · Acceptance letter]

25 Oct 2024

PCOMPBIOL-D-24-00678R1 

A Computational Framework for Understanding the Impact of Prior Experiences on Pain Perception and Neuropathic Pain

Dear Dr Ramne,

I am pleased to inform you that your manuscript has been formally accepted for publication in PLOS Computational Biology. Your manuscript is now with our production department and you will be notified of the publication date in due course.

With kind regards,

Anita Estes
